# From Digital Mental Health to Digital Social and Emotional Wellbeing: How Indigenous Community-Based Participatory Research Influenced the Australian Government’s Digital Mental Health Agenda

**DOI:** 10.3390/ijerph18189757

**Published:** 2021-09-16

**Authors:** James Bennett-Levy, Judy Singer, Darlene Rotumah, Sarah Bernays, David Edwards

**Affiliations:** 1University Centre for Rural Health, The University of Sydney, Lismore 2480, Australia; judy.singer@sydney.edu.au (J.S.); d.edwards@sydney.edu.au (D.E.); 2Gnibi College of Indigenous Australian Peoples, Southern Cross University, Lismore 2480, Australia; darlene.rotumah@scu.edu.au; 3School of Public Health, The University of Sydney, Sydney 2006, Australia; sarah.bernays@sydney.edu.au; 4London School of Hygiene and Tropical Medicine, London WC1E 7HT, UK

**Keywords:** community-based participatory research, Indigenous Australians, community partnerships, digital social and emotional wellbeing, digital mental health, Aboriginal and Torres Strait Islander health, Indigenous community engagement, First Nations research

## Abstract

This paper describes the first six years of a government-initiated project to train Indigenous health professionals in digital mental health (d-MH). It illustrates how community-based participatory research (CBPR) methods were used to enable this “top-down” project to be transformed into a ‘ground-up’ community-guided process; and how, in turn, the guidance from the local Indigenous community partners went on to influence the national government’s d-MH agenda. The CBPR partnership between five community partners and a university rural health department is described, with illustrations of how CBPR harnessed the community’s voice in making the project relevant to their wellbeing needs. The local Indigenous community’s involvement led to a number of unexpected outcomes, which impacted locally and nationally. At an early stage, the conceptual framework of the project was changed from d-MH to the culturally-relevant Indigenous framework of digital social and emotional wellbeing (d-SEWB). This led to a significant expansion of the range and type of digital resources; and to other notable outcomes such as successful advocacy for an Aboriginal-specific online therapy program and for a dedicated “one-stop-shop” d-SEWB website, *Wellmob*, which was funded by the Australian government in 2019–2021. Some of the implications of this project for future Indigenous CBPR projects are discussed.

## 1. Introduction

The aim of this paper is to present a case study of the first 6 years (2013–2019) of an Australian Federal Government-funded digital mental health (d-MH) project, which has received recurrent funding since 2013. The purpose of the project has been to design and deliver d-MH training programs for Aboriginal and Torres Strait Islander health professionals (henceforth, respectfully referred to as Indigenous Australians or Indigenous Peoples) and other health professionals working with Indigenous clients to use d-MH resources with their clients. This paper builds on a previous paper, published two years after the start of the project [1], which illustrated how this top-down government-funded d-MH project was transformed into a community-guided ground-up project.

In the present paper, we illustrate the inverse: how the ground-up community-guided processes impacted not only locally but nationally. Further, we illustrate how locally-generated advocacy played a key role, alongside of government flexibility, in creating a far more culturally relevant project. A key element in creating cultural relevance was the re-framing of the project from a sole focus on d-MH to encompassing the Indigenous Australian framework of social and emotional wellbeing (SEWB) [2].

In the following sections we provide the background and context for the Methods and Results under these headings:The e-Mental Health in Practice project (eMHPrac)Digital mental health (d-MH)The Indigenous concept of social and emotional wellbeing (SEWB)Digital social and emotional wellbeing (d-SEWB)Community-based participatory research (CBPR)

### 1.1. The e-Mental Health in Practice Project (eMHPrac)

In 2012, the Australian Federal Government was one of the first governments internationally to launch a national digital mental health strategy, reasoning that digital mental health services were uniquely placed to “reach those people who are currently not accessing services, overcoming issues of distance, cost and stigma” [3]. One element of this strategy, initiated in 2013, was to develop a suite of training programs to train GPs, allied health professionals and Indigenous health professionals to utilize digital mental health (d-MH) resources in their practice. Over the past eight years, a consortium of four Australian universities, led by Queensland University of Technology (QUT), have collaborated on this project, which became known as eMHPrac (www.emhprac.org.au, accessed 29 June 2021).

Three of these universities from different regions of Australia (Queensland, Northern Territory and northern New South Wales), were responsible for designing and delivering digital mental health (d-MH) training for Indigenous health professionals in their respective regions. Due to regional and workforce differences, these training programs were developed independently.

This paper describes some of the outcomes which resulted from developing training program at one of those university centres, the University Centre for Rural Health, a rural department of The University of Sydney, based on Bundjalung country in Lismore, Northern New South Wales. The UCRH training programs, called “*R U Appy*” and “*Ur Mobile: a tool 4 Wellbeing*” are described in the Methods section.

### 1.2. Digital Mental Health (d-MH)

Digital mental health refers to services targeting mental health problems via online and mobile technologies. At the start of this project, the conventional understanding about d-MH (or e-mental health as it was then known) was that it was limited to evidence-based online therapy programs [4,5,6]. Compared with most nations, Australia was well-advanced, having already developed and researched a range of online programs prior to 2010 [7,8,9]. Typically, these programs consisted of several online modules, with or without therapist support, based on cognitive behaviour therapy principles. At that time, it was the Australian Government’s understanding that d-MH was essentially synonymous with evidence-based online therapy programs. However, this left a gap for Indigenous Australians as no culturally adapted online therapy programs had been developed.

Mental health apps for computing devices including tablets and smartphones were still in the early phases. With considerable foresight, one culturally adapted app for Indigenous Australians, *Stay Strong*, had been developed for iPad use by our eMHPrac partners from the Menzies School of Health Research [10,11]. In 2013, this was the only culturally adapted d-MH tool available for Indigenous Australians, except for a pilot version of a suicide prevention app for tablets, *iBobbly*. This pilot version was made available to our project by our project partners, the Black Dog Institute, before becoming more widely available for use on smartphones, tablets, and iPad—for further details see [11,12].

### 1.3. The Indigenous Concept of Social and Emotional Wellbeing

A key element in the evolution of the University Centre for Rural Health training program was a re-conceptualisation of digital mental health to digital social and emotional wellbeing. As we reported in a previous paper [1], this shift was instigated by the project’s community partners in the early phase of the project. In the present paper, we provide greater detail on the key role of the community partners in creating this shift. Furthermore, one of our aims here is to illustrate how this shift led to a series of unexpected outcomes, instigated by the community partners, that were not envisaged by the government funders or anticipated by the academic researchers.

The term SEWB has come to be synonymous with an Indigenous Australian understanding of a holistic approach to health and wellbeing [2,13,14]. Over the last 20 years a growing academic literature, as well as numerous landmark SEWB reports, have reclaimed and given primacy to the culturally-rich Indigenous conceptions of wellbeing that are typically aligned to a strengths-based approach as opposed to a deficit narrative around health and wellbeing [13,14,15,16,17,18].

Although Indigenous peoples’ understanding and application of SEWB varies across the different contexts within Australian urban, rural and remote Indigenous communities, the following definition [2] provides a well-recognised explanation: “SEWB is a multidimensional concept of health that includes mental health, but which also encompasses domains of health and wellbeing such as connection to land or ‘country’, culture, spirituality, ancestry, family and community” (p55). Central to the SEWB model is the understanding of the impact of colonisation on the wellbeing of Indigenous Australians through political, historical, social and cultural factors [13].

### 1.4. Digital Social and Emotional Wellbeing (d-SEWB)

As will be described in this paper, the term digital social and emotional wellbeing (d-SEWB) is a concept that specifically emerged in this project through the consultations with Indigenous community partners. Just as the concept of social and emotional wellbeing is much broader than mental health, so the term d-SEWB, which first featured in our 2015 paper [1], relates to different types of digital resources, across a wide range of cultural domains from Indigenous connection to land and sea, community and kin, as well as individual domains of body, mind and emotions. In this context, d-MH is seen as a subset of a much wider range of d-SEWB resources that can contribute to an Indigenous person’s wellbeing.

### 1.5. Community-Based Participatory Research

As we previously reported, community-based participatory research (CBPR) has underpinned the project from its inception [1]. CBPR has been widely recommended by researchers as an appropriate way to undertake research with Indigenous communities [19,20,21,22]. It has been suggested that CBPR “provides the gold standard for equitable, partnered research in traditional communities” (p. 1) [23].

CBPR is a collaborative approach to research that equitably involves community members, organisational representatives, researchers and others in all elements of a research project. The aim of CBPR is “to enhance understanding of a given phenomenon and the social and cultural dynamics of the community, and integrate the knowledge gained with action to improve the health and well-being of community members” (p. 177) [24].

CBPR recognises the fundamental importance of involving members of the researched communities as active and equal participants in all phases of a research project. In the context of the present research, “community” meant involving Indigenous people living on Bundjalung country in northern New South Wales. This included: community Elders, Indigenous health and community professionals, and community members of all ages and experiences, who might have personal or family experiences with mental health problems; and also younger community members who typically had more experience in using digital devices. The project team has included Indigenous and non-Indigenous members throughout its various stages.

In a CBPR project, partnerships should be collaborative and equitable, and promote co-learning and capacity building [19,20,22,24,25]. As we shall describe, valuing Indigenous guidance and building local Indigenous capacity from the outset were important elements in enhancing community confidence in the project. CBPR projects should also involve a iterative process with ideas and actions cycling between community and researchers [20]. Fortunately, the project has been funded over an extended period of time, leading to multiple iterations of project methodology and outcomes as guided by CBPR partners.

Other key elements of CBPR processes have been described in a number of publications [19,26,27,28]. Evaluation is important to ensure the integrity of these processes. Previously, we evaluated our CBPR practices against *“10 principles for research with Aboriginal and Torres Strait Islander peoples”* [29], and found them to be consistent with all but one of these principles [1]. The exception at the start of the project was Principle 1, *“addressing a priority health issue as determined by the community”* [20,29]. While the Federal Government had determined in 2013 that training Aboriginal health professionals in d-MH was a national priority, this was not a priority as far as the local Bundjalung community were concerned at the start of the project. However, as we illustrate in the present paper, the extensive engagement and commitment by the community partners was central in turning this top-down project into a ground-up community-guided project—a project that then made sense to the local Indigenous community and was also applicable to the diverse communities across the country.

### 1.6. Aims of the Paper

The context for this paper is the transformation of a top-down government funded d-MH training project to a community-guided, ground-up d-SEWB training project, fuelled by CBPR processes; and a description and analysis of how these CBPR processes led to significant outcomes at both local and national levels. Within this context, the primary aims of this paper are to:Provide concrete illustrations of the range of CBPR approaches used.Describe some of the key local and national project outcomes, and how these were linked to the CBPR processes. These links are illustrated in the Results section.Draw out some of the key implications for projects within Indigenous communities: in particular, implications for transforming top-down governments projects, maximising community involvement, providing adequate funding and timelines, building trusting relationships leading to co-learning, building Indigenous capacity and leadership, and prioritising time for team reflection.Make visible the contributions of the community partners to the intellectual, ethical and cultural integrity of the project.

## 2. Materials and Methods

This section sets the context for the CBPR processes and project outcomes reported in the Results section. First, we briefly identify the five Community Partners. Their roles and contributions are described in much greater detail in the Results section. Second, we describe the data collection methods and analysis (see Table 1). Third, we identify the University Centre for Rural Health team and the authors of this paper. Fourth, we provide details of the two training programs, *R U Appy* and *Ur Mobile: A Tool 4 Wellbeing*, and the *WellMob* d-SEWB “one-stop shop” website. Impacts on these training programs and on the development of the *Wellmob* website were some of the key outcomes from the CBPR processes (see Results section).

Written meeting notes from each AG. Circulated to all AG members as part of iterative process. Included as Appendices in reports to Dept of Health. Transcribed interviews with 2 AG community leaders by author JS.

### 2.1. The CBPR Partners

Of the five Community Partners, three were responsible for *project governance*: the Ngayundi Aboriginal Health Council, Indigenous project Advisory Groups and the Aboriginal Health & Medical Research Council Ethics Committee. The other two partners, the Indigenous Learning Circles and Indigenous Workshop Participants based on Bundjalung country, had a different role. They provided feedback on their direct experience of d-SEWB resources. A full description of the Community Partners’ roles and processes are reported in the Results section.

### 2.2. Data Collection Methods and Analysis

As indicated in Table 1, the data for this paper are derived from a range of sources over the duration of this CBPR project:Meeting notes from the Ngayundi Aboriginal Health CouncilMeeting notes from the Advisory Group and interviews with two members who were community leadersWritten feedback from the AH&MRC Ethics CommitteeTranscripts and thematic analysis of audio recordings of the Learning Circle discussionsInterviews with Workshop Trainers and their written reportsInterviews with Workshop Participants

Most of the interviews, transcriptions and data analysis were undertaken by author JS with credibility checks by author JB-L. Author DR undertook interviews, transcriptions and data analysis for one of the studies [31] together with the lead author for that study, Jennifer Bird. All interviews were semi-structured, based on interview schedules approved by AH&MRC ethics committees. Thematic analysis of the data see [30,31] followed standard qualitative methods described by Braun and Clarke [32].

The project received ethics approval and approved amendments from the Aboriginal Health & Medical Research Council (NSW) AH&MRC-HREC (955/13) and the Northern NSW Local Health District NCNSW-HREC (076).

### 2.3. The University Centre for Rural Health Team

Between 2013 and 2019, the project team employed 10 staff: 6 Indigenous, 4 non-Indigenous; 7 female, 3 male. These 10 staff undertook a variety of roles, many in combination: for instance, community engagement, program development, delivering training, program evaluation, research, coordination, mentoring, support and administration. Four of the staff had been employed on the project for more than three years (four of the authors); other staff were employed for between six months and two years.

All the Indigenous staff were well known and respected members of the local Indigenous communities. Their expertise and connections across the different sectors were instrumental in building stronger links between the University and community and enabled the project to develop more relevance within these Indigenous communities.

Two of the present authors are Indigenous. DE is a Worimi man who started as a member of AG in 2013. In 2017, he was employed on the project, first as a trainer and later as Director of the Wellmob project. DR is a Bundjalung woman who played a key role from 2013–2016 in the formative community engagement processes and in the evaluation of the first training program.

The other three authors are non-Indigenous researchers. Two have been with the project since its inception in roles as project director (JBL) and researcher/facilitator (JS). SB has not been employed in the project, but has contributed specialist advice on conceptualization, methodology and data analysis.

### 2.4. The d-SEWB Training Programs and the WellMob Website

Table 2 outlines the contents of the two digital SEWB training programs, *R U Appy* (2014–2016) and *Ur Mobile: A Tool 4 Wellbeing* (2017–2019). As reported in the Results section, advice and feedback from the five CBPR partners was central to the design of these programs. This feedback also inspired the advocacy for the development of the *WellMob* website (2020–ongoing)—see Table 2.

## 3. Results

The Results are divided into two main sections. In the first section, we describe the roles of the community partners and illustrate the processes that were created to enable them to have maximum involvement in the design and implementation of the project.

In the second section, we highlight five areas where the guidance and feedback from the community partners has had direct and significant impact on the outcomes of the project for Indigenous health and community professionals (HCPs) and their clients.

### 3.1. The Roles and Contributions of the Community Partners in Supporting the Research

Table 3 and Table 4 provide a description of the Community Partners and the different roles they played across the duration of the project. Three of the Community Partners—gayundi Aboriginal Health Council, the project’s Advisory Groups and the AH&MRC Ethics Committtee—were central to the governance of the project (see Table 3). Their feedback and advice helped to steer the project in all kinds of helpful ways, thereby avoiding many of the pitfalls that may undermine the pertinence of projects to the community. For instance, as described in an earlier publication [1], the willingness of the Ngayundi Aboriginal Health Council to engage with the researchers enabled a government ‘top-down’ project to be transformed into a ground-up, community-endorsed project.

The other two Community Partners—Learning Circle participants and Workshop Participants—provided valuable feedback and advice which helped to shape the structure and content of the training programs and the development of the *WellMob* website (see Table 4 and the outcomes described in Section 3.2).

Coupled with Figure 1, these Tables indicate the extent to which the Community Partners were integral to the process and outcomes of the project.

Facilitating a process in which engagement with our five research partners could be meaningful was approached very deliberately throughout the project, with an emphasis on the continuity of their involvement. Figure 1 provides a visual summary of the number of meetings or sessions attended by the five Community Partners. This figure clearly illustrates the regularity and extent of their involvements over the six-year period covered by this paper. Considerable effort and resources were invested in scheduling regular sessions, using familiar and appropriate meeting structures and techniques, to enable partners to influence the evolution of the project.

As will be shown in the next section describing the impact of these partnerships, their ongoing involvement was reflected in a series of responsive adaptations, in which the feedback from partners led to direct changes and shifts in the project’s implementation. The contributions of the three governance groups in Table 3 and the two groups which directly experienced the d-SEWB resources (Table 4) appeared to operate in a cyclically reinforcing manner, further augmenting the partners’ trust and investment in the project.

The project partnership structure was developed to support the effective engagement of each partner group to play a key role in steering the approach and direction of the project. For example, the overarching guidance provided by the Ngayundi Aboriginal Health Council enabled the project to be tailored to become more directly relevant to the community’s context. The Indigenous Advisory Groups ensured that the appropriate cultural protocols and procedures were in place throughout the project. The influential voices of the lived experiences of those providing, needing and receiving mental health and SEWB were represented by the Indigenous Advisory groups and within the Learning Circles and training workshops. Finally, the AH&MRC ethics committee ensured that feasible and appropriate community engagement strategies, which emphasised the pertinence and value of the project to the Indigenous community, were adopted and implemented.

### 3.2. Impacts of the Community Partnerships on the Design and Implementation of the Project

Below we highlight five areas where the community partnerships had significant impacts on the outcomes of the project for Indigenous health and community professionals (HCPs) and their clients.

The five areas are:Changing the conceptual framework of the project from digital mental health (d-MH) to digital social and emotional wellbeing (d-SEWB)Expanding the range and types of digital resources to encompass Indigenous SEWB domainsSuccessfully advocating for an Indigenous-specific online therapy programProviding key inputs into the development and design of the *R U Appy* and *Ur Mobile for Wellbeing* training programsSuccessfully advocating for the development of what is now the *WellMob* website, the first Indigenous-specific “one-stop shop” portal of d-SEWB resources.

#### 3.2.1. Changing the Conceptual Framework of the Project from Digital Mental Health (d-MH) to Digital Social and Emotional Wellbeing (d-SEWB)

In the early stages of the project, the Ngayundi Aboriginal Health Council and the Advisory Groups (AGs) suggested that the concept “digital mental health” was problematic and recommended changing to “digital social and emotional wellbeing.” As an AG member remarked: “*Aboriginal people look at things from more of a holistic perspective, [we] need the flexibility to include multiple areas of a person’s life*”.

In isolation the term “mental health” was seen to be culturally inappropriate as it did not reflect a holistic approach to wellbeing that included for instance, connection to culture, kin and country [2]. For the training to be relevant and appealing to Indigenous HCPs, it needed to be embedded within a SEWB framework. As noted by a HCP in the LCs, *“we want to listen to ourselves, and we want to hear ourselves reflected in other [Indigenous] people that [makes the digital space] culturally appropriate”.*

In response to the Ngayundi and AGs feedback, the team’s second bi-annual Progress Report to the Department of Health in December 2014 advocated for a community-endorsed conceptual shift from d-MH to d-SEWB. The Department accepted this report, laying the groundwork for broadening the d-SEWB resource base in the following years, and for the future development of the Indigenous d-SEWB website, *WellMob*.

#### 3.2.2. Expanding the Range and Types of Digital Resources to Encompass the SEWB Domains

While the Ngayundi Health Council and AGs had identified the need to reframe the project from d-MH to d-SEWB, it was HCP participant feedback from the 2015 training program that pinpointed the need to include a far wider range of digital resources to reflect work role diversity [30].

The training participants emphasized that Indigenous HCPs have a wide variety of work roles and qualifications (e.g., health promotion, support roles, crisis support, drug and alcohol counsellors, Aboriginal education officers etc.). What these HCPs required were digital resources that were culturally appropriate and fitted within their scopes of practice and competencies [30,31]

It became clear that widening the range of resources also fitted much better with the holistic concept of SEWB and the broader health advocacy roles of HCPs. The HCP workshop participants and 2016 Learning Circle feedback indicated that the HCPs wanted culturally appropriate digital resources that related to all aspects of improving the SEWB of their clients and not just deal with symptomatic health and wellbeing issues. As explained by one HCP*, “we are so multifaceted in our ways of being and our spirituality, we’re not going to fit into a box like ‘oh you have alcoholism, just treat that’. Treat the things around it and you might treat the alcoholism”.*

The cultural integrity of d-SEWB resources was a central theme. One LC participant remarked that d-resources *“need to be in language that our communities understand”.* This could include videos in language about connecting to country, d-resources to assist healing for Stolen Generations, or d-resources that celebrated Indigenous culture. The key point expressed by HCPs was that the d-resources needed to be culturally specific: *“I really like the [d-resources] made by Indigenous people, I like hearing [Indigenous] people’s stories around suicide and mental health issues and how the person has come through it, it makes it relatable”.*

A wide range of d-resources were understood to promote SEWB, yet technically they sat outside of the narrow definition of a “mental health” resource. Accordingly, this broader definition of d-SEWB informed the 2017 *Ur Mobile: A Tool 4 Wellbeing* program and was central to the development of the *WellMob* website.

#### 3.2.3. Successfully Advocating for an Aboriginal-Specific Online Therapy Program

As noted above, the community partners advocated for the holistic SEWB framework to underpin the digital training strategy. However, they also recognised and strongly endorsed the need for evidence-based mental health treatments, delivered in culturally appropriate ways.

This was especially acknowledged by the 2014 Learning Circles who recognised that online therapy programs, supported by qualified health professionals, had a greater capacity than apps to deliver a full program of psychoeducation and strategies for change. However, in 2014, there were no Indigenous-adapted online therapy programs. This lack of any cultural adaptation of existing online programs was seen as a major barrier to their engagement and use by Indigenous Australians. All the programs were perceived to have been designed for the dominant Caucasian culture.

There was general agreement with one LC participant’s comment about a non-Indigenous online program that it “wasn’t very inviting, it was drab to read, like you might as well have read a government website or something, that’s how drab it was”. Another participant noted that “it was just whitefellas stories … they didn’t look like people I could relate to.”

The Learning Circles’ strong recommendation was that evidence-based online therapy programs should be adapted for Indigenous Australians. The project team promoted this recommendation to *Mindspot*, a national online therapy provider that had been funded by the Federal Government as part of its d-MH strategy [6]. *Mindspot* readily agreed to the request, and by the following year, *Mindspot* had created Australia’s first—and still only—culturally adapted online therapy program for Indigenous clients [33].

#### 3.2.4. Providing Key Inputs into the Development and Design of the *R U Appy* and *Ur Mobile: A Tool 4 Wellbeing* Training Programs

Central to the development of the *R U Appy* and *Ur Mobile for Wellbeing* training programs was the advice and feedback from Ngayundi, the AGs, the Learning Circles and workshop participants.

At an early stage, Ngayundi Aboriginal Health Council and the AGs drew the project’s attention to the number of Aboriginal HCPs who were over 40 and might have limited technology skills. This observation was reinforced by the younger members of the Learning Circles who emphatically recommended training the older workforce *“to be up to date and savvy with technology before using the apps [with clients].”* Specific modules to upskill workers in the use of devices (iPads, tablets, smartphones, etc.) prior to engaging with the d-SEWB resources were included from the start of *R U Appy* training and progressively refined.

Ngayundi and the AGs also pointed out that there was considerable variation in the level of counselling skills amongst Indigenous HCPs. In the early phases of the project, the *R U Appy* program featured training in the use of the *Stay Strong app* [33], at that stage the only Indigenous digital SEWB app available. *Stay Strong* required basic counselling skills for effective use by HCPs, so counselling modules were therefore included in the *R U Appy* program.

Understanding the needs of the local workforce led Ngayundi, the AGs and Learning Circles to advocate for post-training follow up support. Therefore, at the completion of the *R U Appy* program, five months of monthly small group supervision “booster” sessions were offered and accepted by 28 Indigenous HCPs [30]. The HCPs’ feedback highlighted the need to expand the range of SEWB resources to fit their varied roles. In the booster sessions, HCPs were more interested in searching for Indigenous-specific wellbeing and health education resources, rather than focusing on mental health resources.

Since there was no central location to find Indigenous-specific digital resources, the feedback from these supervision groups led directly to the next iteration of the training program, the 2017–2019 *Ur Mobile: A Tool 4 Wellbeing* program. In this new half to full day workshop*,* the focus was on developing the skills necessary to search and save Indigenous-specific d-resources to create a personalised digital library relevant to the HCP’s particular role.

#### 3.2.5. Successfully Advocating for Wellmob, a ‘One Stop Shop’ SEWB Website

The frustrations experienced by *R U Appy* and *Ur Mobile 4 Wellbeing* training participants and by members of the Learning Circles were paramount in enhancing advocacy for a dedicated “one stop shop” SEWB website for Indigenous people. *Ur Mobile: A Tool 4 Wellbeing* participants and Learning Circle members were confronted to discover that searching the internet for Indigenous specific and culturally relevant SEWB resources to HCP work was *“like looking for needles in haystacks”.* Although by 2017 there were more Indigenous d-resources (videos, apps, etc.) available on the Internet than at the start of the project, they were still difficult to find. A common response from the Learning Circles is captured by this member, *“I came to the understanding that these resources aren’t easy to find [searching] feels like [hitting] another brick wall and realising there’s nothing there we need to put them [resources] in a space where we can find them”.*

Consistent with the concept of SEWB, the kinds of d-resources that Learning Circle participants and training participants were seeking extended far beyond a narrow conception of mental health. Learning Circle participants reported that they particularly valued resources made by and for Indigenous peoples, which featured culture and language. What was clearly apparent to all participants was the need for a central hub where all relevant SEWB resources for Indigenous people could be found.

Informed by this feedback, the project team’s advocacy for a dedicated SEWB website grew stronger and more persistent from 2016. Supported by other colleagues in the broader eMHPrac consortium, representations to the Department of Health were finally successful, resulting in funding for the development of the *WellMob* website in 2019–2021. *WellMob* was launched in July 2020 after developing through a similar CBPR approach. At the time of writing, we believe *WellMob* to be the first dedicated ‘one-stop shop’ digital SEWB website for Indigenous peoples anywhere in the world.

## 4. Discussion

Eight years (to date) of recurrent funding for a government-funded project is a rarity, albeit via a series of 1-to-2-year project renewals. It offers an important lens through which to evaluate the value of community guidance for Indigenous projects. It also creates the opportunity to reflect, re-evaluate and modify research and program directions as part of an iterative process across CBPR partners. As we have described, from the outset CBPR processes have been central to the d-SEWB training program of the project team. Previously, we have reported that the first two years of this project were largely devoted to transforming the government’s agenda to train Indigenous health professionals in d-MH into a community driven ground-up project [1]. The present paper charts the project’s first six years. We have illustrated its local and national impacts during that period. In essence, the infusion of CBPR processes has created a U-shaped project over this time: from top-down to ground-up in the first 12–18 months; and then from a ground-up local Indigenous community base to shifting the government’s d-MH agenda towards d-SEWB for all Indigenous Australians.

Below we discuss a range of topics emerging from this case study: Creating alignment of project goals across community, researchers and government; addressing d-MH/d-SEWB issues of equity for Indigenous peoples; the role of CBPR processes in Indigenous projects; six learning outcomes emerging from these years of engagement; study limitations; and conclusions.

### 4.1. Creating Alignment of Project Goals across Community, Researchers and Government

The government agenda for this project had been to develop and deliver a d-MH training program for Indigenous health professionals and other non-Indigenous health professionals working with Indigenous clients. Outcomes were to be measured in the number of training programs and the number of health professionals attending these programs. For the local Bundjalung Indigenous communities, the primary agenda was not the number of programs, but their design and value to their community. Fortunately, the government accepted the project team’s proposal that allocating adequate time for local community input should be central to the design and outcomes of the training programs.

### 4.2. Addressing d-MH/d-SEWB Issues of Equity for Indigenous Peoples

From the outset, it was clear that there was a lack of Indigenous-specific d-MH resources. Even after the project’s focus had moved from d-MH to d-SEWB, one of the key learnings was the difficulty Indigenous Learning Circle members and workshop participants had in locating Indigenous d-SEWB resources—“like looking for needles in haystacks”. Their advocacy for a one-stop shop and the subsequent development of the *WellMob* website has made a big difference to resource visibility, accessibility and potential to contribute to Indigenous Peoples’ wellbeing.

*WellMob* provides a portal for over 240 psychoeducational resources such as videos, web-links, and apps that can contribute to health education, cultural identity and connections to community and country. However, there remains a lack of culturally appropriate evidence-based programs for Indigenous Australians with mental health disorders. There is an issue of equity here, since the lack of such resources for Indigenous peoples both in Australia [34,35] and internationally [36,37] stands in stark contrast to the preponderance of evidence-based online therapy programs for majority cultures [37,38] and the huge number of mainstream mental health apps [39]. At this stage, 8 years after the start of the project, there is one evidence-based online therapy program developed for Indigenous Australians [33] and two culturally relevant mental health apps that have a limited evidence base [11,12,34]. This lack of d-MH resources with an evidence base remains a problem as there is significant and growing need [36,40], particularly amongst young Indigenous peoples [35].

### 4.3. The Role of Community-Based Participatory Research (CBPR) in Indigenous Projects

In this paper, we have sought to illustrate the impacts of five different community partnerships on project outcomes. We have described how these different partnerships influenced different outcomes at different stages in the project’s development. Our collaborative partnership goal has been to create a d-SEWB training program that would be of tangible benefit for Indigenous HCPs and their clients. That goal is ongoing. We have learned as much through the shortcomings of the *R U Appy* and *Ur Mobile: A Tool 4 Wellbeing* programs [1,30] as through the successes [31]. The iterative reflective processes of the learning circles and workshop participants’ feedback taught us that non-Indigenous d-MH programs were culturally inappropriate; that more Indigenous-specific d-SEWB and d-MH resources were needed; and that there was a clear need for a one-stop-shop d-SEWB website.

Part of authentic CBPR is not pre-determining the direction or the outcome of the community’s input. Here what we wish to emphasize is that these outcomes were unexpected. The project team did not set out to create those outcomes, and they would not have been anticipated by government. The outcomes were emergent from CBPR processes; their impact was to create a far more culturally relevant and dynamic approach to enhancing Indigenous mental health and wellbeing. The reconceptualization of the project from d-MH to d-SEWB has expanded the range of digital resources seen to be appropriate for training Indigenous HCPs; has led to the development of *WellMob*; and led to the advocacy for the evidence-based Indigenous online therapy program developed by *Mindspot*. We hope that this project provides a useful window into the responsive agenda of CBPR when it is afforded adequate time, commitment and recognition of its value.

### 4.4. Six Learning Outcomes

Here we highlight six key learning outcomes emerging from this project.

#### 4.4.1. Community Involvement Is Central to Effective Outcomes

First, in line with other researchers, we suggest that a commitment to CBPR practices is the most effective—and perhaps the only—way to achieve outcomes that are truly meaningful and relevant for Indigenous communities [27,41,42,43]. CBPR is an inherently de-colonising research approach, in direct opposition to the kind of ‘top-down’ hierarchical research practices that have previously institutionalised cultural racism [16].

In this project, each of the CBPR partners made unique contributions. Their roles had different emphases. For example, the Ngayundi Aboriginal Health Council asked penetrating ‘big picture’ questions about the value and relevance of the project for the community at the start of the project. The AGs provided important guidance for engaging HCPs in the learning circles and training programs. The AH&MRC Ethics Committee gave ethics oversight and fine-tuned the community engagement processes. The Indigenous Learning Circles were deeply engaged with the digital materials and made crucial contributions to the framing, content and delivery of the HCP training programs. The HCP workshop participants provided valuable feedback about digital resources, which furthered the evolution of the training program and, in combination with the Learning Circles’ findings, led to the development of the *WellMob* website. As other studies have found, it is usually important to have representation of key stakeholders across all project levels from governance to end users, as different skills are needed for different purposes at different points in time in a project [27,41].

The CBPR processes provided the platform which enabled the community to exercise a degree of control over the direction of the research; a shared alignment to the goals of the project; and real outcomes for Indigenous communities across the country. These processes and the community partners’ dedicated commitment to improving the wellbeing of all Indigenous people led them to put aside their initial scepticism and further investigate the potential digital resources could play in a person’s wellbeing.

#### 4.4.2. Providing Adequate Funding and Timelines Enables Meaningful Community Input

As we have previously emphasised, Indigenous projects of this nature require sufficient time and funding to allow for meaningful community input [1,28,44]. Consequently, a key recommendation is that adequate time and funding for community consultations needs to be built into the design and length of projects. We recognise that this project has been privileged to receive 8 years of funding to date. This has allowed us—and the community—to witness the accumulated benefits over time, as knowledge, understanding and the relationships between CBPR partners deepen.

It is not feasible for all projects to receive this level of support. However, we suggest that short-term project funding (e.g., 6 months to 2 years) is often a futile exercise for Indigenous projects, especially those led by non-Indigenous researchers. The initial funding was for three years. That allowed for a meaningful assessment of effectiveness on the part of the funders and enabled a strong platform to be built for the project’s future in the event of further funding. There is an important cost-benefit issue here. In isolation, giving time and resources to strong community engagement might be considered a high additional cost. However, if this investment makes the difference between effective and ineffective outcomes, then it is more than justified. Future studies should aim to assess the health economic benefits of this relationship between community involvement and project outcomes.

#### 4.4.3. Set out to Create Co-Learning Hubs

Third, and relatedly, it takes time to build the kind of trusting relationships and understanding which lead to creative thinking and co-learning—one of the hallmarks of best practice CBPR [20,27]. When a community-academic partnership creates co-learning hubs where community and researchers recognise their complementary expertise, they are open to learning from one another [27]. In this project, the community partners’ knowledge and understanding of the local community, Indigenous SEWB perspectives and the HCP workforces were central to the design of the d-SEWB training programs; while the researchers understood the broader context of the government-funded project, knew of available d-SEWB resources and facilitated the research processes. Co-learning was intrinsic to the process. For instance, although the researchers had a good level of knowledge and relative expertise about digital resources, the Learning Circles and workshop participants’ feedback were invaluable in understanding how they were experienced by community members and HCPs of different ages, roles and experience.

#### 4.4.4. Project Outcomes Are Greatly Enhanced by Building Local Capacity

Fourth, the project employed Indigenous staff, who were well known to many of the Bundjalung community representatives. The project also embedded local community leadership through its governance structures (Ngayundi Aboriginal Health Council and Advisory Groups) and promotion of staff into leadership positions. This was particularly important as two of the core team members were non-Indigenous, albeit with prior links with Indigenous services, community leaders and HCPs. Local capacity building was further embedded in the project through the Learning Circles and training programs. All these developments served to enhance the Indigenous community’s confidence in the project, build trust, and enable important conversations between the project team and the community [1].

#### 4.4.5. Time for Reflection Needs to Be Intrinsic to CBPR Processes

Fifth, the team prioritised time for reflection. Reflection is acknowledged to be a key part of CBPR methods such as participatory action research [14,20,28,45]. Our practice was that team members wrote reflective notes or research memos after each significant event and held regular team meetings in which team members could explore their experiences in depth. As an example, the data for one of our publications were the written reports and verbal reflections of the two R U Appy trainers who conducted follow-up “booster” sessions with supervision groups [30]. The present paper is a further example, borne of the realisation that it is important to deliver a reflective publication which foregrounds the potential of CBPR processes to deliver meaningful outcomes. We concur with Lin et al. [28] who concluded in their review of community engagement approaches for Indigenous research that “it is essential for researchers to engage in a continuous process of self-reflection throughout all research stages” (p. 10).

#### 4.4.6. Advocacy Is an Important Component of CBPR Processes

Another key features of CBPR is that it can be a powerful tool for policy advocacy [46]. This project attests to the importance of giving voice to the community’s needs through different media e.g., qualitative research, reports to government, conference presentations, community forums, and direct advocacy to stakeholders. CBPR’s advocacy role is important in promoting Indigenous wellbeing while ever there are embedded disadvantages in the history and resourcing of services for Indigenous Australians.

### 4.5. Study Limitations

There are a number of limitations inherent in this study. It would have been valuable to have the CBPR partners evaluate which aspects of the CBPR process they found most helpful or unhelpful [27]. However, government funding for the project was geared towards the development of d-MH/d-SEWB training; therefore, the formal evaluation of the project was principally oriented towards training outcomes. Since this was a CBPR process, the project outcomes described in this paper were emergent rather than prescribed, developed through an iterative process. Consequently, outcomes from CBPR were to a large extent different from those required by government. Whereas the CBPR process was focused on making the training program as culturally relevant as possible, government reporting requirements were more focused on the numbers of training programs and numbers of HCPs attending.

It could be argued that another limitation was the amount of community time and resource that was required in the various CBPR processes. We are well aware that Indigenous communities have many demands and are often being over-burdened with consultation. This project has been fortunate in being so well supported. Whether or not similar results could have been achieved without so much consultation is uncertain. To transform the project from “top down” to “community-guided” did require a number of meetings of Ngayundi Aboriginal Health Council and the Indigenous Advisory Groups, especially in the first two years of the project (see Figure 1). This enabled the community partners and the project team to actively work together to design a training program that the community could feel satisfied would meet their needs. Learning Circle members were paid as consultants for their cultural knowledge and engagement with the d-MH and d-SEWB resources. They attended groups for 15 to 18 h and provided important feedback. To be at their most effective, CBPR processes may require considerable resourcing. This is perhaps a further learning outcome and recommendation from this project.

COVID-19 certainly enhanced interest in digital health and wellbeing resources and a recognition of the specific needs of Indigenous peoples [34,35,47]. It would be pleasing to report that culturally relevant evidence-based d-resources and training programs are in place and ready to support health promotion and low intensity mental health interventions. The development of resources such as the *WellMob* SEWB website, the *Mindspot Wellbeing* program and apps such as *Stay Strong* and *iBobbly* are steps in the right direction. However, there is still a long way to go, not only to develop many more appropriate d-resources, but also to establish how best to integrate them into work practice in health and community settings [30,31,48,49].

## 5. Conclusions

In summary, our aim in this paper has been to demonstrate how CBPR processes, embedded in a long-term project, have fundamentally altered its trajectory, thereby generating significant learnings for the community and university partners, and the government funders. A suite of culturally relevant d-SEWB resources, have been brought together on the *WellMob* website and are now available for d-SEWB training programs, is the result. It is extremely unlikely that these changes could have been delivered through a simple “top down” process. However, in this instance, we were fortunate that our funders at the Department of Health recognised the importance of community engagement; and were sufficiently flexible to take on board and fund the community’s recommendation for a dedicated d-SEWB website.

Lastly the project has immense gratitude for our community Elders and Advisory Groups, who could easily have dismissed the government’s “top-down” imposition of a d-MH project on the community. Instead, they chose to work with us, asking the hard and important questions that were needed to transform the project into one that would be meaningful to their communities and, in turn, to Indigenous communities across the country. Throughout the process, the Elders and Advisory Groups and HCPs gave generously of their time and wisdom. In writing this paper, one of our aims has been to make explicit the extent and depth of the intellectual, ethical, and cultural contributions of all Aboriginal and Torres Strait people involved in this project.

## Figures and Tables

**Figure 1 ijerph-18-09757-f001:**
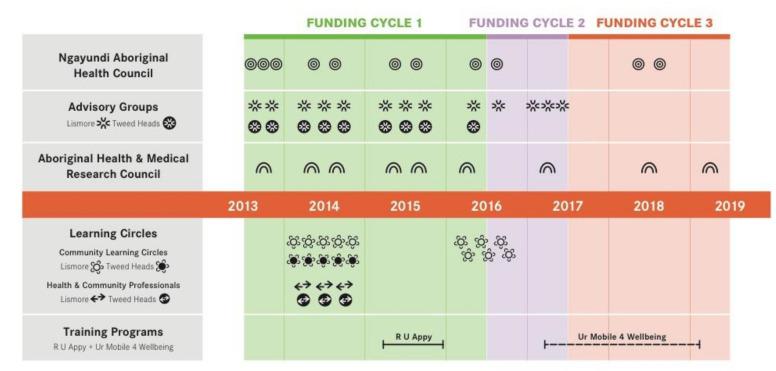
Number of meetings or sessions attended by the five community partners.

**Table 1 ijerph-18-09757-t001:** Outline of methods.

Community Partners	Years; Numbers of Data Collection Events	Data Management
Ngayundi Aboriginal Health Council	2013–2018 11 meetings	Review of written meeting notes
Advisory Group (AG)	2013–2019 22 meetings	Written meeting notes from each AG. Circulated to all AG members as part of iterative process. Included as Appendices in reports to Dept of Health. Transcribed interviews with 2 AG community leaders by author JS.
AH & MRC Ethics Committee	2013–2019 9 applications	Ethics applications and amendments enabled AH&MRC ongoing review of research, cultural protocols and governance structures.
Learning Circles	2014: 16 sessions/36 participants 2016: 6 sessions/7 participants	Audio recordings for 16 sessions. Transcription and thematic analysis by JS. Key themes in reports to Dept of Health and Mindspot. 6 participant interviews and thematic analysis by JS. Key themes in report to Dept of Health.
Workshop Participants	2015: Interviews with the 2 trainers, and their written reports 2016: Interviews with the 16 participants	Transcription and thematic analysis of interviews and written reports by JS, credibility check by author JB-L. Reported in publication [30]. Transcripts and thematic analysis of interviews by Jennifer Bird and author DR. Reported in publication [31].

**Table 2 ijerph-18-09757-t002:** The University Centre for Rural Health training programs and the *WellMob* website.

R U Appy Training 2015	Ur Mobile: A Tool 4 Wellbeing Training 2017–2019	WellMob Website 2020–Ongoing
**What:** 3-day workshop, plus follow-up support **Where:** Lismore and Tweed Heads **How many:** 5 programs **Program:** **Day 1**—Orientation to digital technologies **Day 2**—Learn about the Stay Strong app **Day 3**—Using *Stay Strong* in practice, including skill devel- opment in micro-counselling, risk assessment and referral skills and goal setting. **Follow-up Support**—Monthly ‘booster’ sessions for 5 months	**What**: Half-day workshop to build resource library, sometimes combined with *Stay Strong* skills building workshop **Where:** NSW **How many:** 13 workshops **Program:** **Half-day:** Build a d-SEWB resource library by developing technology skills to search, assess and save d-SEWB resources in the device of choice Half-day (optional): Using *Stay Strong* in Practice	**What:** Australia’s first ‘one-stop-shop’ website for Indigenous-specific SEWB resources, including videos, apps, podcasts, websites, online programs, social media **Where:** www.wellmob.org.au (accessed on 29 June 2021) **Who:** Indigenous frontline workers and communities, and others working with Indigenous people **Why:** Feedback from Community Partners about the need for a SEWB website ‘made by and for mob’

**Table 3 ijerph-18-09757-t003:** Community partners: governance groups.

**Ngayundi Aboriginal Health Council**	**Who?** Indigenous representatives from across Bundjalung country. Primarily a Community Elders Council, it also involved Indigenous HCPs including CEOs, program managers from Government and NGO services, and was open to all Indigenous people interested to attend. Number of attendees varied from around 20–40 at different meetings. **What?** Regional governance role to review and monitor various health and wellbeing programs proposed for the local community. This included guidance on the project design, research methods and engagement strategies, as well as formal endorsement of the project. **Where?** Meetings rotated around Bundjalung Country, e.g., Tweed Heads, Lismore, Ballina, Casino, Kyogle, Fingal Heads
**Indigenous Advisory Groups** **(AGs)**	**Who?** Included service managers, Elders, Indigenous HCPs, and community members with lived experience. 6–10 members in the 2 AGs—see below. **What?** The project’s “in-house brains trust”, crucial in shaping community involvement strategies; in particular, the recruitment to the Learning Circles and engagement of/with local service managers; made significant contributions to ensuring appropriate cultural protocols were in place; and to the structure, design and content of the training package. **Where?** AGs in 2 locations across the Bundjalung nation: Lismore/Widjabul-Wyabul country and Tweed Heads/Minjungbal country.

**Table 4 ijerph-18-09757-t004:** Community partners: Learning Circles (LCs) and workshop participants.

**Learning** **Circles**	**Who?** Community members and Health and Community Professionals (HCPs) 2014: two cohorts (n = 32): community members (n = 16) and HCPs (n = 16) 2016: one mixed group of community members and HCPs (n = 7) **What?** Purpose: to enable an immersive experience over time in order to: (1) user-test and provide feedback on a variety of d-SEWB resources (2) to provide advice about the structure, design and cultural appropriateness of the *R U Appy* and *Ur Mobile for Wellbeing* d-SEWB training programs. **Where?** 2 locations, Lismore, Tweed Heads **How often?** 2014: three-hour LC sessions for Community Members ran over 5 consecutive weeks, across the two locations (15 hrs total); the LC for HCPs were over 3 weeks due to work commitments (9 h). 2016: three-hour LC sessions over 6 weeks for mixed group ran in one location, Lismore (18 h).
**Workshop Participants**	**Who?** Over 300 Indigenous and non-Indigenous HCPs attended the *R U Appy* and *Ur Mobile: A Tool 4 Wellbeing* programs. Detailed feedback was obtained from 28 Indigenous HCPs who attended “booster sessions” following the *R U Appy* 3-day training program (2015). **What?** Report and analysis of participants’ feedback are described in previous papers [30,31] **Where?** 6 booster groups across 4 regions on Bundjalung Country **How often?** Up to 5 booster sessions over 5 months

## Data Availability

Data are available upon request. Data underlying our findings cannot be made public for ethical reasons, as they contain information that could compromise the privacy and consent of research participants. Data requests may be sent to the corresponding author.

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
