# Peer review of "From Digital Mental Health to Digital Social and Emotional Wellbeing: How Indigenous Community-Based Participatory Research Influenced the Australian Government’s Digital Mental Health Agenda"

_ijerph, 2021, doi:10.3390/ijerph18189757_

Round 1

Reviewer 1 Report

Please see below my responses:

Briefly summarize the content of the manuscript;

The manuscript describes how a “top-down” project of training indigenous health professionals in digital mental health was converted to a “ground-up” project through community-based participatory research.  The outcomes had both local and national impacts.  The team had used the involvement of both Indigenous and non-indigenous members all throughout the various stages of the project. 

2. Illustrate what are, in your opinion, the manuscript’s strengths and
weaknesses [this is an essential step, because the Editor will consider
the reasoning behind your recommendation and needs to understand it
properly];

In my opinion, the major strength of this manuscript is getting first-hand outcomes directly from the people concerned and not general statements or expectations from another person’s viewpoints.  Another aspect is the concept of taking the digitalized mental wellbeing practices to the Indigenous people.

3. Provide a point-by-point list of your major recommendations for the
improvement of the manuscript;

The paper by itself is quite a good study and is also well-written and I don’t have any major improvements to suggest.

4. If necessary, provide a point-by-point list of your minor
recommendations for the improvement of the manuscript.

None.

Author Response

Thank you very much for your endorsement of this paper. We agree that the outcomes are the greatest strength of the paper. In response to Reviewer 2's comments, we have now strengthened the reporting of the Methods section providing greater detail. We hope that this further strengthens the paper. We are attached a revised version (clean version)

Reviewer 2 Report

This manuscript "From digital mental health to digital social and emotional wellbeing: How Indigenous community-based participatory research influenced the Australian government’s digital mental health agenda" would be of great interest to the readership of the International Journal of Environmental Research and Public Health. The authors describe a case study of a project to train Indigenous health professionals in digital mental health. The authors describe how community-based participatory research (CBPR) methods were used to transform the project into a community-guided process. This paper could make an important contribution to the literature; however, several issues would need to be addressed before it is ready for publication.

My main concern with this paper is the lack of detail provided for the case study’s methods. Much of the information included in the “Materials and Methods” are, in fact, results of the case study, particularly Tables 1 and 2 and Figure 1, and should be moved to the results section. It is not until the very end of the “Materials and Methods” section that we are provided with bullet points on the sources of data provided in this case study; yet much more information should be provided. How many interviews were conducted with each group? Who conducted the interviews? Were the interviews structured or unstructured? Were they recorded and transcribed? How were the data analyzed?   

The following are additional (mostly minor) suggestions to improve the manuscript:

  1. I would suggest defining terms much more explicitly throughout the paper. Although the authors clearly define the term “social and emotional wellbeing,” the paper could be improved by clearly defining other terms for the reader (e.g., CBPR, digital mental health).
  2. At the end of the third paragraph of the introduction, please note that abbreviations should be added in the bullet points to be consistent with how they appear in the subsequent subheadings.
  3. Table 3 is introduced to the reader at the end of the sixth paragraph of the introduction, prior to any other table being introduced. I would suggest deleting this reference to Table 3 in the introduction and referring to tables in the narrative only in the order in which they are numbered.
  4. The introduction has a subsection titled “Digital Mental Health (d-MH). In the last sentence of this subsection, the authors refer to tool called iBobbly. A sentence or two might be added here to explain this tool for the reader.
  5. I would suggest adding text to walk the readers through the figure and tables. Providing a summary and analysis of each would help the reader better understand and make sense of the information in the figure and tables. In addition, it is unclear why information provided for some groups in Tables 1 and 2 is more detailed than others. For example, we are presented with exact numbers of individuals participating in the Learning Circles, but not for the Ngayundi Aboriginal Health Council. The data source for this information, such as sign-in sheets, also should be described in the method section.

Author Response

Thank you very much for your review. We have really appreciated its thoroughness and the points you have made. In particular, you noted weaknesses in the Methods section, and suggested moving the Tables in the Methods section to the Results. Please see our point-by-point response to your review below. We thank you for these suggestions. We have made changes in accordance with them. These have now strengthened the paper.

We have uploaded a revised version of the paper as an attachment with track changes still in place so you can see how we've responded (we understand that this revised changes and a clean version can only be formally uploaded once you have responded here, but it seemed easiest to upload now so you can see the changes). 

Below are our responses to your suggestions:

Point 1: My main concern with this paper is the lack of detail provided for the case study’s methods. Much of the information included in the “Materials and Methods” are, in fact, results of the case study, particularly Tables 1 and 2 and Figure 1, and should be moved to the results section. It is not until the very end of the “Materials and Methods” section that we are provided with bullet points on the sources of data provided in this case study; yet much more information should be provided. How many interviews were conducted with each group? Who conducted the interviews? Were the interviews structured or unstructured? Were they recorded and transcribed? How were the data analyzed?   

Response to Point 1: 

We have now addressed all these points in a revised Methods section. In particular, we have included much greater detail about the methods and included a new Table 1 entitled Outline of Methods. This provides details of the number of data collection events over time, and the data management, including the nature of the data, interviews, recording, transcriptions, who carried them out, methods of data analysis, and previous published papers [30, 31] which have provided greater detail about the data analysis. The manuscript adds to the details provided in Table 1.

The previous Tables 1 and 2, and Figure 1 have now been moved to Results. The Results section now have two parts: the first part which includes Tables 1 and 2 and Figure 1 discusses the process in much greater detail under the subheading "The Roles and Contributions of the Community Partners in Supporting the Research."

The second part discusses Outcomes as previously.

Point 2:

The following are additional (mostly minor) suggestions to improve the manuscript: I would suggest defining terms much more explicitly throughout the paper. Although the authors clearly define the term “social and emotional wellbeing,” the paper could be improved by clearly defining other terms for the reader (e.g., CBPR, digital mental health).

Response: We have now clearly defined the terms CBPR and digital mental health - see attachment p.2, lines 35-36 and p.3, lines 44-49

Point 3: At the end of the third paragraph of the introduction, please note that abbreviations should be added in the bullet points to be consistent with how they appear in the subsequent subheadings.

Thank you, we have now addressed this by adding the abbreviations

Point 4: Table 3 is introduced to the reader at the end of the sixth paragraph of the introduction, prior to any other table being introduced. I would suggest deleting this reference to Table 3 in the introduction and referring to tables in the narrative only in the order in which they are numbered.

Response: Thank you, this reference has been removed from the Introduction

Point 5: The introduction has a subsection titled “Digital Mental Health (d-MH). In the last sentence of this subsection, the authors refer to tool called iBobbly. A sentence or two might be added here to explain this tool for the reader.

Response: We have now added a further sentence explaining the tool as "a suicide prevention app for tablets" in the pilot version. Then adding with references: "This pilot version was made available to our project by our project partners, the Black Dog Institute, before becoming more widely available for use on smartphones, tablets, and iPad - for further details, see [11, 12]." - see page 2, lines 52-56

Point 6: I would suggest adding text to walk the readers through the figure and tables. Providing a summary and analysis of each would help the reader better understand and make sense of the information in the figure and tables. In addition, it is unclear why information provided for some groups in Tables 1 and 2 is more detailed than others. For example, we are presented with exact numbers of individuals participating in the Learning Circles, but not for the Ngayundi Aboriginal Health Council. The data source for this information, such as sign-in sheets, also should be described in the method section.

Response: We have added 3 paragraphs of new text at the start of the Results section under the heading "The Roles and Contributions of the Community Partners in Supporting the Research." We hope this better helps to guide the reader in relation to the Figure 1 and Tables 3 and 4 (previously 1 and 2).  We have also fleshed out Tables 3 and 4 in some places to better explain the roles of the community partners.

We have added numbers ("around 20-40 attendees") for Ngayundi meetings. Numbers for these community meetings were very variable. We have no access to these records, and indeed only attend for a limited time in each half-day meeting to present our project. Numbers for other community partners we can be clearer about because these groups were facilitated by the project team.

Round 2

Reviewer 2 Report

Thank you for addressing all of the reviewer comments.